# Water, Sanitation and Hygiene in Schools in Low- and Middle-Income Countries: A Systematic Review and Implications for the COVID-19 Pandemic

**DOI:** 10.3390/ijerph19053124

**Published:** 2022-03-07

**Authors:** Kasandra I. H. M. Poague, Justine I. Blanford, Carmen Anthonj

**Affiliations:** Faculty of Geo-Information Science and Earth Observation–ITC, University of Twente, Hengelosestraat 99, P.O. Box 217, 7500 AE Enschede, The Netherlands; j.i.blanford@utwente.nl (J.I.B.); c.anthonj@utwente.nl (C.A.)

**Keywords:** water supply, WASH, hand disinfection, handwashing, menstrual hygiene management, SARS-CoV-2, developing countries, students, education, human rights

## Abstract

The global COVID-19 pandemic has revealed the extent to which schools are struggling with the provision of safe drinking water, sanitation and hygiene (WASH). To describe the WASH conditions in schools and discuss the implications for the safe reopening of schools during the ongoing COVID-19 pandemic, a systematic review of peer-reviewed literature on WASH in schools in low- and middle-income countries was performed. In April 2021, five databases, including MEDLINE (via PubMed), Web of Science, Scopus, AJOL, and LILACS, were used to identify studies. Sixty-five papers met the inclusion criteria. We extracted and analyzed data considering the Joint Monitoring Programme (JMP) definitions and the normative contents of Human Rights to safe drinking water and sanitation. Publications included in this systematic review considered 18,465 schools, across 30 different countries. Results indicate a lack of adequate WASH conditions and menstrual hygiene management requirements in all countries. The largely insufficient and inadequate school infrastructure hampers students to practice healthy hygiene habits and handwashing in particular. In the context of the COVID-19 pandemic, being hindered to implement such a key strategy to contain the spread of SARS-CoV-2 in the school environment is of major concern.

## 1. Introduction

The transition between the years of 2019 and 2020 was marked by the emergence of several cases of atypical pneumonia in Wuhan, China, which would be later confirmed as cases of Severe Acute Respiratory Syndrome (SARS) caused by a novel coronavirus [1]. The SARS-CoV-2 virus, as the etiologic agent was named, associated with Coronavirus Diseases 2019 (COVID-19), rapidly spread throughout the world despite numerous national and international efforts aimed at its containment [1]. On 11 March 2020, the World Health Organization (WHO) declared the COVID-19 outbreak caused by the SARS-CoV-2 a pandemic [2]. As of 10 January 2022, there have been 305,914,601 global confirmed cases of COVID-19, including 5,486,304 deaths, reported to WHO and a total of 9,126,987,353 vaccine doses against COVID-19 have been administered [2]. 

Although it initially appeared that young children were less susceptible to SARS-CoV-2, and generally when infected with COVID-19 they had a relatively mild course, the number of children and young people admitted to hospital during the pandemic has increased with the emergence of new variants [3,4,5,6]. Despite respective media reports [5,6], evidence indicating that new variants of coronavirus may spread more easily in children is lacking. Infected children experience the same symptoms as adults (fever, cough, vomiting, diarrhea, sore throat, dyspnea), with gastrointestinal symptoms being more frequent when compared to other age groups [3,4]. Moreover, an increasing number of complete or incomplete Kawasaki disease and Multisystem inflammatory syndrome cases in children (MIS-C) were reported in various countries during SARS-CoV-2 epidemics, suggesting an association between COVID-19 and these diseases [7,8,9,10]. 

The role children play in the transmission and spread of COVID-19 is still ambiguous. What is known is that children have played an important role in past disease outbreaks and epidemics [3,11]. Taking into account that most of the children infected with COVID-19 are asymptomatic or usually develop mild symptoms, the virus may remain undetected in children and young people, turning them into potential superspreaders of SARS-CoV-2 [3,10]. Furthermore, children have difficulties in describing minor symptoms related to COVID-19 (e.g., myalgia, headache, anosmia and ageusia). Along with the similarity of clinical manifestations of COVID-19 to other pediatric infectious diseases, SARS-CoV-2 in children can be easily misdiagnosed [12]. Faced with this problem, based on evidence of the benefits of school closures from former influenza outbreaks and as an attempt to contain the spread of the COVID-19 pandemic, most governments around the world have temporarily closed educational institutions [13].

The United Nations International Children’s Emergency Fund (UNICEF) estimates that the imposed countrywide school closures in 188 nations during the pandemic affected over 1.6 billion students so far [14]. School closures entail several adverse impacts, not only on children, but also on other stakeholders (parents, teachers, school staff, etc.) [15]. Some of the downsides include the increase in school evasion and inequalities across the globe once access to digital technologies and, therefore, to the new mode of e-learning is unequal [14]. On top of the accumulative adverse effects of the school closure, its effectivity is uncertain. A systematic review conducted by Viner et al. [16] found no evidence of the relative contribution of school closures to SARS-CoV-2 spread containment. Therefore, for children to get back to “normal” life and learning, especially with the progress in COVID-19 vaccination, the reopening of schools has already started, with most of the schools around the world being partially open. 

In order to support educational institutions during school reopening phases, the World Health Organization (WHO) released a checklist of protective measures, which include and acknowledge the provision of safe drinking water, sanitation and hygiene (WASH) as essential to ensure adherence to hand hygiene etiquette and further COVID-19 prevention in schools [17]. The absence of adequate WASH infrastructures, such as handwashing stations, water, and soap, hampers the practice of handwashing, which is one of the fundamental strategies to contain the spread of the virus. Furthermore, the presence of the virus and its genetic material in feces of COVID-19 patients [18,19] and in sewage [20,21,22] suggests that fecal-oral transmission may serve as an alternative infection route for SARS-CoV-2. Several studies indicate that SARS-CoV-2 can remain viable for days in raw sewage, sewage sludge, surface water and feces of patients [23]. Hence, the inadequate disposition of wastewater could lead to the contamination of water sources and surfaces, resulting in the fecal-oral transmission of COVID-19.

The role that the provision of safe WASH services in schools plays in reducing infectious disease exposure and transmission (including helminth infections, diarrhea, respiratory and other communicable diseases) has been important prior to the COVID-19 pandemic [24]. Likewise, safe WASH in schools promotes innumerable health and educational benefits such as: (i) reducing school absence by providing an appealing learning environment for children [25]; (ii) boosting student’s cognitive skills and consequent performance by preventing dehydration [26]; (iii) promoting a clean, healthy and secure environment for menstrual hygiene management for girls [27,28,29]; (iv) promoting WASH education by introducing students to the concepts of drinking water, water-related diseases and environmental health-related topics [30]; and (v) influencing hygiene practices and encouraging behavior change in the students’ families and community, once children also act as agents of change outside the school environment [31]. 

Moreover, access to safe drinking water and sanitation is formally recognized by the United Nations as a human right. Its provision in all settings, including schools, should follow the normative contents of availability, accessibility, affordability, quality and safety, acceptability, privacy and dignity [32,33]. However, as pointed out by WHO and UNICEF through the WHO/UNICEF Joint Monitoring Programme (JMP), worldwide and especially in low-and middle-income countries (LMICs), the majority of schools lack the necessary WASH infrastructure to ensure the safety of the school community during the school reopening [34]. Despite the recognition of hand hygiene as a key aspect to prevent COVID-19, hitherto WASH interventions are rarely included in the list of structural and environmental measures carried out in the schools to reduce transmission of SARS-CoV-2 [35,36].

To address the insufficient WASH infrastructure in schools in LMICs and consequent implications for students’ health, firstly, however, research is needed to identify and describe WASH conditions in LMICs during the COVID-19 pandemic. In order to fill this knowledge gap and also to highlight the importance of WASH in schools, we conducted the first descriptive systematic review on WASH in schools in LMICs. We sought to understand:(i)What is the situation of water, sanitation and hygiene conditions in schools in LMICs?(ii)What are the implications of the current WASH conditions in schools in LMICs for the safe reopening of schools during the ongoing COVID-19 pandemic and for future water-related pandemics?

## 2. Materials and Methods

### 2.1. Search Strategy

This review and analysis of studies reporting WASH conditions in schools were conducted in adherence with the guidelines of the Preferred Reporting Items for Systematic reviews and Meta-Analyses (PRISMA) [37]. In April 2021, the following databases were systematically searched, adapting search terms according to the requirements of each database: MEDLINE (via PubMed), Web of Science, Scopus, African Journals Online (AJOL) and Literatura Latino-Americana e do Caribe em Ciências da Saúde (LILACS). Search terms were divided into three blocks—Block 1 (water, sanitation and hygiene), Block 2 (schools and students), Block 3 (low-and middle-income countries). Blocks were combined using boolean operator AND while search terms within the blocks were combined using boolean operator OR. Medical Subject Headings (MeSH) and Descritores em Ciências da Saúde (DeCS) were exclusively included in searches conducted in MEDLINE (via PubMed) and LILACs, respectively. For LILACS, the research was carried out with the same descriptors of the other databases in English, Portuguese and Spanish. DeCS terms equivalents to the MeSH terms were used in Portuguese and Spanish. Search terms were translated to Portuguese and Spanish where applicable. As there is not an advanced search function on the AJOL website, for this specific database, the search was conducted in Google Scholar, restricting the search results from AJOL website. The search terms were adapted to the 256 characters limitation of Google Search, and only the first 1000 results were included due to Google’s export limitation. The detailed search strategies applied for different databases are available in the Appendix A. Reference lists of included studies were hand-searched for additional publications not identified through databases. Prior to the screening of titles and abstracts, this systematic review was registered in PROSPERO (CRD42021248831).

### 2.2. Selection Criteria

Titles and abstracts of each publication were screened and checked against the inclusion criteria for a full-text review. Studies published in English, Portuguese or Spanish were considered. There was no publication date restriction. Any article that presented a description of WASH conditions in schools in low- and middle-income countries, regardless of its design, outcome and aim, was eligible for inclusion. Regarding the educational level, the review considered studies investigating early childhood education, primary education and secondary education institutions regardless of the population targeted (students, teachers, parents, school staff, etc.). In the case of several studies containing the same data by the same group of authors, the study with the most detailed description was included and the other studies were excluded. Any literature meeting one or more of the criteria shown in Table 1 was excluded.

### 2.3. Data Extraction

Following the search, all identified references were collected and uploaded into Rayyan free web tool [38] for systematic reviews, and duplicates were removed. Information on water and sanitation conditions in schools was extracted and analyzed considering the five normative contents of the human right to water and sanitation (HRTWS), namely [32,33]:(i)Availability: water supply must be sufficient and continuous, and sanitation facilities should be available for use at all times of day and night in sufficient numbers.(ii)Accessibility: water and sanitation facilities should be physically accessible within or in the immediate vicinity of the environment to all at all times. The design of the facilities should also take into account elderly people, young children, and persons with disabilities.(iii)Affordability: water and sanitation services must be affordable for all. The costs must not affect peoples’ capacity to secure other essential necessities guaranteed by human rights.(iv)Quality and safety: The water must be safe, therefore free from micro-organisms, chemical substances, and radiological hazards that constitute a threat to health (i.e., it should follow the national, local, or international guidelines for drinking water quality). As for sanitation, the facilities must be situated where physical security can be safeguarded and must be hygienic. Wastewater and excreta must be safely disposed to effectively prevent human, animal, and insect contact with human feces, and the infrastructure should be constructed to prevent collapse.(v)Acceptability, privacy, and dignity: All water and sanitation facilities and services must be culturally appropriate and sensitive to gender, lifecycle, and privacy requirements. Water organoleptic properties such as odor, taste, and color should be acceptable, and sanitation facilities must have their design, positioning, and conditions of use sensitive to people’s cultures and priorities. That includes gender-separated facilities, infrastructure that ensures privacy, and appropriate resources for menstrual.

In addition, water source and sanitation facilities in schools were classified as improved or unimproved according to the definitions of the JMP [34]. Improved water sources include piped water, boreholes or tubewells, protected dug wells, protected springs, rainwater, and packaged or delivered water. Unimproved sources include unprotected wells, unprotected springs and surface water. Improved sanitation facilities include flush/pour-flush toilets, ventilated improved pit latrines, composting toilets and pit latrines with a slab or platform. Unimproved sanitation facilities include pit latrines without a slab or platform, hanging latrines and bucket latrines. An overall lack of water and sanitation in schools was grouped with the unimproved category. A new category named “Unknown” was created for cases in which the description of the water source or sanitation facility was provided in the study, however, could not be classified as improved or unimproved, such as: (i) the water source or type of sanitation facility was not listed in the JMP definitions; (ii) the study did not describe the water source or sanitation facility of all the schools assessed but rather just part of it; (iii) the study described the water source and sanitation facility in general, however, the number of schools with the correspondent water source or sanitation facility was not provided. Even though hygiene is indirectly addressed in most of the normative contents, the HRTWS focuses exclusively on water and sanitation. Hygiene infrastructure in schools was evaluated considering the definitions of the JMP service ladders for WASH in schools [34]. These definitions classify school hygiene according to the simultaneous presence of handwashing facilities, water and soap into basic, limited, or with no hygiene service. Due to the limitation of information from studies, however, classification of schools according to the JMP service ladders was not possible. It is worth noticing that data extraction and analysis had to be constantly adapted due to the lack of data, standardization of information and definitions adopted across different studies. Information on school hygiene infrastructure was later compared with the WHO checklist of protective measures in schools [17].

Data on MHM was extracted and assessed considering three of the five basic resources and preconditions defined by the WHO and UNICEF that must be met to ensure that teenage girls will be able to take care of menstrual-related needs while at school [39], namely: (i) access to menstrual hygiene materials to absorb or collect menstrual blood; (ii) access to facilities that provide privacy for changing materials and washing of the body with soap and water; (iii) access to disposal facilitates for used menstrual materials (from collection point to final disposal). Table 2 summarizes the topics covered and the extracted data. 

### 2.4. Quality Assessment

Study quality was assessed using the Mixed Methods Appraisal Tool (MMAT) [40], a critical appraisal tool for systematic mixed studies reviews, i.e., reviews that include qualitative, quantitative and mixed methods studies. Papers were classified into five categories: (i) qualitative (ethnography, phenomenology, narrative research, grounded theory, case study or qualitative description); (ii) quantitative randomized controlled trials; (iii) quantitative non-randomized (non-randomized controlled trial, cohort, case-control, and cross-sectional analytic study); (iv) quantitative descriptive (incidence or prevalence study without comparison group, survey, case series and case report); (v) and mixed methods researches (convergent design, sequential explanatory design and sequential exploratory design). The definition proposed by MMAT for cross-sectional analytic studies was expanded to include not only health outcomes, but also behaviors (e.g., hygiene and menstrual hygiene management practices) and school-related outcomes (school absence) as well.

The MMAT tool provides a checklist with seven questions (two general screening questions for all papers and five specific questions for each study design) and a brief explanation of each query, the study categories and some key references for the classification of studies [40]. The first two questions investigate whether the research questions are clear and whether the methodology used in the study is appropriate to address the questions. For qualitative studies, the next five questions assess the coherence between the methodology used, the findings, the conclusions presented in the studies and whether the qualitative approach is the most adequate for the research objective. As for quantitative randomized controlled trials, the questions focus on the randomization performed in the study, characteristics of control and intervention groups, adherence to the intervention and blinding. The questions for quantitative non-randomized studies and quantitative descriptive studies revolve around the representativeness of the target population, sampling strategy, whether measures and statistical methods are appropriate, whether confounding factors have been taken into account, and completeness of the data. For mixed methods studies, 17 questions need to be addressed (two general screening questions, plus the first five for qualitative research, five for quantitative randomized controlled trials, or quantitative non-randomized, or quantitative descriptive and five for mixed methods studies). The last five questions explore the integration and coherence of the qualitative and quantitative methods employed. 

For each paper, the seven questions were rated as “Yes”, “No”, and “Can’t tell”. The results are presented as percentages based on the number of criteria met (100% when all five specific criteria were met and 0% when none of the criteria were met). Studies that failed to meet the MMAT criteria for screening were not scored, yet were still included in the analysis since the quality assessment was not used as an exclusion criteria. 

## 3. Results

### 3.1. Search Results

In total, 6287 articles were identified by searching the five databases (2061 from MEDLINE via PubMed, 2518 from Scopus, 1000 from AJOL, 682 from Web of Science, and 76 from LILACS). After removing duplicates, the titles and abstracts of the 4168 remaining studies were screened and 119 were identified as eligible for full-text review. Forty papers were included in the initial systematic review. All included papers were hand-searched for additional bibliographical references, resulting in an additional twenty-five papers to be included. Finally, 65 papers were included in this review (Figure 1).

### 3.2. Study Characteristics

All included papers were published in English and covered a wide range of topics (Table 3). The studies took place in 30 different countries, most of them located in Africa (53%, n = 16), Asia (33%, n = 10) and less frequently in Central (10%, n = 3) and South America (3%, n = 1). Eight percent of the studies (n = 5) were conducted in multiple locations (countries) while the other 92% (n = 61) focused on only one country. Ethiopia, Nigeria and Uganda were the countries where WASH in schools was more frequently assessed (in 17%, n = 11; 11%, n = 7; and 12%, n = 8 of studies, respectively). The location where studies were conducted, frequency of studies per country and the theme addressed in the papers are presented in Figure 2. 

The majority (51%) of the studies were conducted exclusively in primary or elementary schools (n = 33), followed by the combination of primary and other levels (20%, n = 13), only secondary schools (12%, n = 8) and junior, high and preparatory schools (8%, n = 5). Ninety percent of studies (n = 6) identified the grades and ages of the pupils rather than the school level. The number of schools in studies varied between a minimum of 1 to a maximum of 10,000. The results of this systematic review refer to a total of 18,465 schools. Sixty-nine percent of studies (n = 45) provided information about the locality of school (5502 schools located in rural sites vs. 633 urban schools) or management model (1259 public or government-run schools vs. 484 private schools). More information about the studies such as year of data collection, number of schools per locality (rural vs. urban) or management model (public vs. private) can be found in the Appendix A.

### 3.3. Quality Assessment Results

The 65 studies included in the systematic review comprised a wide variety of study designs (38%, n = 25 quantitative descriptive studies; 22%, n = 14 qualitative studies; 20%, n = 13 quantitative non-randomized studies; 14%, n = 9 mixed methods studies; and 6%, n = 4 randomized controlled trials). Five percent of the papers (n = 3 qualitative studies) failed to meet the MMAT criteria for screening [59,74,98] and, therefore, were not scored. Of the remaining 62 studies, in general, most of the papers (74%) were scored with high percentages (52%, n = 32 studies were assigned with 100% and 23%, n = 14 studies were ranked with 80% score). Qualitative and mixed methods studies had the highest scores, while randomized controlled trials had the lowest (half of the papers scored 40% and the other half 20%). The quality assessment scores for all studies are presented in more detail in the Appendix A.

### 3.4. Water

#### 3.4.1. Availability

Ninety-eight percent of the studies (n = 60) addressed access to water in schools. However, only half of these studies described the type of water source. Information on water sources was provided in total for 16,963 schools (details can be found in the Appendix A). Figure 3 summarizes the classification of water sources in schools as improved, unimproved or nonexistent, and unknown according to the location of the study settings. Considering all settings, 66% of schools had an improved water source. Unknown sources included public tap, standpipe, piped water from a spring, handpump and groundwater (wells, pumped wells, shallow well water, open well and ordinary dug wells). One study informed that the schools had improved water sources, however, did not define what was considered as improved or describe the sources in detail [46]. Four schools in the Makoko area (Nigeria) and one school in Vhembe District (South Africa) had no water source on school premises and, therefore, had to resort to purchasing water from a private borehole or packaged sachet water popularly known as “pure” water [47,96]. Although the water source was not located on the school premises, according to the definitions of the JMP they were also classified as an improved water source. 

Water shortage (intermittent or no water supply) in schools was reported in 62% of papers (n = 40) (see Appendix A). Water consumption was less than 5 L per capita per day in most of the schools assessed in Dessie City (Ethiopia) [45] and equivalent to 0.6 L per capita per day in schools of Mareko district (Ethiopia) [94]. To meet their water needs during classes, schoolchildren used to bring water from their own home or from other sources (e.g., buy in the market or fetch water from nearby houses) [62,63,71,78,80,82,100]. The opposite situation was also reported by Ngwenya et al. (2018) in the Ngamiland district (Botswana) [87]. Families living near the schools also drew water from the school storage tanks when the school gates were unlocked. The longest period reported of schools without water was two consecutive weeks [87]. To deal with the irregularities in the water supply, schools also resorted to an alternative water source such as a back-up reservoir [45], alternative water source without description, [72], storing water from the main source in tanks [62,87,96] or limiting the access to the water to selected times of the day [29]. Four studies stated the ratio of water tap to school population across schools, which ranged on average from 1:7 to 1:114 [50,69,77,99]. 

#### 3.4.2. Accessibility

Twenty percent of the studies (n = 13) entailed location or distance from the water source in schools. Fourteen percent of the studies (n = 9) reported that the water source was located on premises [27,56,57,62,65,77,84,94,99]. Distance from the water source varied from within 50 m of the school [77], more than 30 m from classrooms with some as far as 350 m away [102], within 1 km of the school [66,91] or more than 1 km away from the school [78,83]. The longest distance reported was 3 km from the school [78]. 

#### 3.4.3. Affordability

The affordability content of the human right to water was discussed in 8% (n = 5) of the papers included in the systematic review. Repair and maintenance of the water infrastructure were limited by a lack of financial resources and the unavailability of spare parts to be locally purchased [44]. Schools generally lacked funds or failed to allocate part of their budget for water (sanitation and hygiene as well) [44,47,69,91]. Reasons for not allocating or not having an exclusive budget for water or WASH included not considering it a priority [91], the costs [91] and limitation of resources [47]. In the four schools of the Makoko slum community (Nigeria), resources were collected through fees paid by students. The amount, which barely covers the operation and maintenance costs of the schools, is not enough to include water provision in the schools [47]. Provision of government’s financial aid to the schools for water was only mentioned by Karon et al. [72] in Indonesian schools. The schools receive a school maintenance and operational grant, designated as BOS, for non-personnel-related operating costs such as water infrastructure repair, water vendor purchase and water treatment [72].

#### 3.4.4. Quality and Safety

Water quality was assessed in schools in 18% (n = 12) of studies across nine countries (Pakistan, Ethiopia, Taiwan, Kenya, Nicaragua, Mozambique, Rwanda, Uganda and Zambia), with 33% of schools (1003 out of 3065) reporting that the water was treated before consumption or chlorine residual was identified in the water. The treatment comprised different methods such as chlorination [41,66,71,91], filtration [41,71], boiling [41,55], disinfection [45], ultraviolet [41], reverse osmosis [55] and solar water disinfection (SODIS) [71], and treatment without specification of methods [84]. None of the schools assessed in Eswatini [78], Cambodia [80] and Nepal [95] provided treated drinking water to students.

Of the seven studies assessing water quality in schools, all detected contamination of water samples with microbial content [41,55,57,80,84,92,95]. According to the *E. coli* counts, 40% of schools (1142 out of 2861) across 15 countries had drinking water in non-conformity with the WHO guidelines (intermediate, high or very high risk) [57,80,84]. More information is presented in Table 4**.**

Problems with maintenance of the water supply was indicated in 18% of studies (n = 12) and comprised reported breakdown of equipment [78] such as pumps [62,99], tube well [80], containers and taps [91], sinks [98], breakdown of the primary water source [84], lack of general maintenance without specification [45,100], existence of non-functional infrastructure [49,71,78,80,87,99], water leakage [98], disconnection and obstruction (clogged or blocked) of water storage tanks [87]. 

#### 3.4.5. Acceptability, Dignity and Privacy

Acceptability, dignity and privacy were content of the human right to water less cited in studies (6%, n = 4). Concerning acceptability, students in the Makoko area (Nigeria) complained about the color, taste and salinity problems of the water sources available in the schools [47]. The lack of disability-friendly designed and accessible water sources for the youngest children in schools was pointed out in 5% of studies (n = 3) [64,72,102]. According to Erhard et al. [64] and Zaunda et al. [102], the current design of water supply with hand-pumps fails to provide enough space for the movement of wheelchairs, and the pump is also not within reach from a wheelchair. Hence, it hampers children with disabilities from having access to water without assistance. In addition, water facilities are also not located in the shade, therefore jeopardizing students with albinism [102]. 

### 3.5. Sanitation

#### 3.5.1. Availability

Eighty-eight percent of the papers (n = 57) evaluated the sanitation services in schools. Sixty-four percent of the studies (n = 41) described the type of sanitation facility found in the schools and 46% of the studies (n = 30) informed the number of students per sanitation facility (more information can be found in the Appendix A). In total, sanitation facilities were assessed in 16,960 schools. Figure 4 summarizes the classification of schools with improved, unimproved or without, and unknown sanitation facilities according to the location of studies setting. Considering all settings, 31% of schools had improved sanitation facilities. Unknown sanitation facilities included latrines, pit latrines without further description, ordinary pit latrines with and without cover, traditional pit latrines, toilets, sanplat latrine, pit, pit latrine with cement floor, pit latrine without cement floor and composting latrine. One study informed that all schools had latrine facilities in the compound that were not adequate, however, lacked a definition of what was considered as adequate or inadequate [94]. In addition to the sanitation facilities, some schools also had urinals, predominantly for boys [43,44,102], although also separated urinals for girls, teachers [43] and shared ones for all students [76]. However, the number of urinals was also small and not enough for the students [102].

Regarding the number of students (boys and girls), only girls, and only boys per sanitation facilities, Figure 5 presents the distribution of the ratios according to the location of studies. Most of the schools failed to meet the national requirements of students per sanitation facilities presented in Table 5**.**

#### 3.5.2. Accessibility

Accessibility to sanitation facilities was assessed in 12% (n = 8) of the included studies.

Sanitation facilities were typically located within the school compound [46], in a secluded area within the school [101], near classrooms [62,102] or close to the school area [62]. The longest distance reported from the classroom to the sanitation facilities was 114 m [102]. Although more than half of the schools assessed in Ibadan (Nigeria) [62] have their toilets located no further than 20 m from the classrooms, 12% (n = 5) and 5% (n = 2) of the schools have toilets located within a distance ranging from 41–60 m and 81–100 m from the classrooms, respectively. The paths for the facilities were not easily accessible [49,102], especially after rains [28,101], were not well maintained [49], dirty [54] and had several barriers hampering children with disabilities to enter or reach (e.g., lack of supporting rails, steps or stairs at the entrance or in the pathways to the facility and rocky pathways) [49,64,102]. Nevertheless, four out of five junior high schools in Kumbungu (Ghana) had toilet facilities easily accessible [82]. 

#### 3.5.3. Affordability

The affordability content of the human right to sanitation was discussed in 12% (n = 8) of the papers included in the systematic review. Financial resources for maintenance, repair or provision of sanitation facilities were provided by non-governmental organizations [43], government [53,72], community [52,53] and families of students [52,71,102]. Schools lacked funding or separate budget for sanitation services [44,52,53,71,101].

#### 3.5.4. Quality and Safety

Sixty percent of the studies (n = 49) discussed the quality and safety of the sanitation facilities in schools in LMICs. Lack of cleanliness of the sanitation facilities was the most frequent complaint mentioned across 51% of studies (n = 33). Facilities were not regularly clean even when there was a cleaning schedule [28,53,60,62,65,71,78,80,87,102], were smelly [44,56,60,78,80,95,101], had visible feces or urine on the floor, walls, seat or premises [52,64,80,95,102] and presence of flies [64,72,95,101,102]. In 100%, 98% and 70% of schools in Trapeang Chour Commune (Cambodia) [80], Ibadan (Nigeria) [62] and Rumphi (Malawi) [102], respectively, the students were the ones responsible for cleaning the toilets. In the case of Rumphi (Malawi) [102], this practice was only registered in public schools. 

With respect to problems with the infrastructure, sewage management and disposal was reported as inadequate in 8% of studies (n = 5), including full toilets and latrines with no subsequent measures taken [44,57], evidence of open drainage and stagnant pools around the school premises [61,76] and blockage of the sewage line [70]. Facilities were also damaged, cracked, broken or had collapsed completely [68,95,99], had poor infrastructure, such as no cement floor, or were built only for temporary use [85], needed repair [62] and put the safety of the learners at risk [87]. There were also reported cases of vandalism [52,62,98], offensive statements on the walls of toilets [70] and in one school in the district of Katakwi (Uganda) some of the latrines had been destroyed by animals [89].

#### 3.5.5. Acceptability, Dignity and Privacy

The content of acceptability, dignity and privacy was discussed in 51% (n = 33) of the studies. Regardless of the existence of sanitation infrastructure, open defecation and indiscriminate urination in the school compound or surroundings were frequent practices [61,62,77,87,98,99,101]. Students reported not using the facilities due to their unsanitary state [54,72,78,101] and a lack of water supply in the facilities [70,72,80]. Fifty-five percent of the students that were interviewed in five schools in Dessie City (Ethiopia) declared that they did not use the toilet and water point correctly. The facilities were often found to be abandoned [63,71,81]. Nevertheless, the majority of the students from the schools in Mereb-Leke District [46], in Dessie City [45], and Mareko District (Ethiopia) [94] reported defecating inside the facilities in the schools. 

Students, especially girls, struggled with the lack of enough privacy in the sanitation facilities in the schools [28,54,92,95]. Use of shared facilities and urinals between students (boys and girls) and between students and teachers was mentioned in 34% (n = 22) and 5% (n = 3) of the studies, respectively. Even when schools had gender-separated toilets, male students were still seen using the female facilities [29]. Schoolgirls also complained about the proximity between boys’, girls’ and teachers’ facilities [28,29]. Lack of doors, locks and roofs in the sanitation facilities or urinals in schools were reported in 20% (n = 13), 17% (n = 11) and 8% (n = 5) of the studies, respectively. Other complaints included the absence of mirrors [49,82], insufficient light [49,56,70], lack of enough space [54,56,82] and ventilation [70]. The facilities and urinals were also without walls [95,102], with holes on the walls [95] or the walls were built of grass or coconut leaves [85]. Moreover, students reported queuing for the use of the sanitation facilities in the schools [46,70,77,94].

The lack of disability-friendly designed and accessible sanitation facilities for the youngest children in schools was pointed out by 5% of studies (n = 3) [64,72,102]. According to Erhard et al. [64], a majority of the latrines in the schools in Malawi had narrow doorways, which hampered students in wheelchairs from entering the facilities without crawling on the floor. In some of the schools of Rumphi town (Malawi), in order to use the toilets, the students had to touch their hands on the ground, which was often dirty [102]. The students with disabilities in schools in Malawi [64] declared they avoided using the latrines during school days, preferring to wait until being back home to do their needs or to practice open defecation, while the students with disabilities from schools in Rumphi town (Malawi) [102] reported reducing their food and liquid intake to decrease their need to use latrines during the school day. 

### 3.6. Hygiene

#### 3.6.1. Availability

Eighty-two percent of the papers (n = 53) assessed the hygiene services at schools. Sixty percent (n = 39) reported lack of soap, 35% (n = 23) mentioned lack of handwashing facilities, 26% (n = 17) indicated lack of water for handwashing, and 25% (n = 16) informed lack of water inside the sanitation facilities in schools. Forty-five percent of studies (n = 29) provided enough information for the quantification of schools with soap (not necessarily in the handwashing facilities), 38% (n = 25) for the quantification of schools with handwashing facilities, 20% (n = 13) for the quantification of sanitation facilities with water and 17% (n = 11) for the quantification of water available for handwashing (see Appendix A). Figure 6 presents the descriptive statistics of hygiene and related facilities in schools according to the studies’ settings.

The average number of handwashing facilities varied from two [43] to five and six [44,92], and the maximum from three [62] to ten [43]. In some schools, in the Sunsari district (Nepal) or Dhaka metropolitan and Manikganj district (Bangladesh), when soap was available it was only designated for teachers [70,90].

Provision of anal cleaning materials (e.g., toilet paper or tissue) was mentioned by 20% of studies (n = 13), which all indicated a lack of self-cleaning materials. Eleven percent of schools (61 out of 559) evaluated reported providing anal cleaning materials for students in Tanzania [44], China [50], Nicaragua [71] and Nigeria [62,65]. Shortage of cleaning supplies for water and sanitation facilities was mentioned in 6% of studies (n = 4) [43,54,80,95]. Schoolgirls had to collect and carry water into the sanitation facilities of the schools in Uganda [56,89].

#### 3.6.2. Accessibility

The location of handwashing stations varied from being near the classrooms [43], on the path from the latrines to the classrooms [80], inside or near the sanitation facilities [43,45,57,72,77], outside the toilet [82], and in the center of schools far away from the sanitation facilities [95,96]. Only one study informed the student-to-handwashing basin ratio as 562:1 [92]. 

#### 3.6.3. Affordability

Shortage of funds and a dedicated budget was the main reason mentioned throughout the papers as the explanation of the lack of soap [59,71,82,90,95]. One of the 55 schools evaluated in Nyanza Province (Kenya) [91] justified the lack of soap by stating it had been stolen from school grounds. Nevertheless, 42% of schools (23 out of 55) in Nyanza Province (Kenya) [91] and 27% of schools (8 out of 30) in Kisumu County (Kenya) were found to allocate funds for soap or hand sanitizer [100]. A lack of funds was also cited as the main reason for the deficit of toilet paper in schools [101]. 

#### 3.6.4. Quality and Safety

Handwashing facilities were described in detail in few studies (only in 6 out of 53) and comprised containers with a tap [43], tippy-taps [43,44], veronica buckets [74,82], a set of two basins being one basin with clean water and a cup to scoop the water over hands and one basin to catch the dirty water [80], concrete rainwater collection structure(s) with tap(s) to turn the water on and off [80], only tap [96] and a big bowl where everybody washed their hands [74].

#### 3.6.5. Acceptability, Dignity and Privacy

The lack of handwashing facilities designed for children with disabilities and for the youngest children to use without assistance was raised by 5% of the studies (n = 3%) [44,64,72].

### 3.7. Menstrual Hygiene Management–MHM

Forty percent of the studies (n = 26) discussed menstrual hygiene management and gender equity in the school environment (for more information, see Appendix A). Lack of access to menstrual hygiene materials to absorb or collect menstrual blood in schools was reported in 15% of papers (n = 10). In addition, 17% of studies (n = 11) mentioned lack of access to facilities that provide privacy for changing, bathing or washing sanitary materials and 20% (n = 13) of studies informed lack of disposal facilities for used menstrual materials such as bins and trash cans for the disposal of sanitary materials. The most frequent method of disposal of menstrual waste was to throw it into sanitation facilities [43,56,60,73,82,87], which led to blockages of the sewage line and latrines filling up quickly. Studies mentioned that the used materials were also not disposed in the schools but rather carried by girls to their homes [43], burned in a rubbish pit [43] and thrown out of toilet windows toward the back of the school compound [70]. Reasons for not using dust bins or other trash receptacles available in the schools included the location of the bins, i.e., bins were located outside of the toilet and therefore were visible for all students [28] or they were too distant from the sanitation facilities and schoolgirls would avoid being seen carrying the materials [56], lack of a management system for transferring used pads to the incinerator or burning [56], and fear of other people (e.g., witch or bad person) or animals to acquire by accident the used materials [97]. Nonetheless, girls reported no problems in the disposal of pads in most of the schools in the district of Katakwi (Uganda) [89].

## 4. Discussion

This systematic review had two main goals. First, to describe the current situation of WASH conditions in schools in LMICs, and second, to understand the implications of these conditions for the safe reopening of schools during the ongoing COVID-19 pandemic, and for future water-related pandemics.

### 4.1. The Current State of WASH Conditions in Schools in LMICs

According to the definitions of the JMP most of the schools evaluated (66%, n = 4842) had an improved water source, while for sanitation, the opposite trend was observed (41% or 6923 schools had unimproved or no sanitation facilities). Although the concept of “improved” water source involves the potential to deliver safe water [34], the classification itself does not guarantee that the drinking water provided by the schools truly safeguards against microbiological and chemical contamination. Problems with water intermittency [47,60,63,87,96] and the contamination of water samples [41,55,95] were reported even in schools with an improved water source. Moreover, packaged and delivered water (e.g., bottle water and tanker truck water) are also considered as improved sources. With no water supply on the premises, four schools in the Makoko area (Nigeria) [47] and one school in Vhembe District (South Africa) [96] had to purchase water from private boreholes and store it in containers or tanks. Although the literature on the quality of water stored in non-household settings such as schools is still scarce, [104] the data from studies on the household level indicate the increase in microbial contamination after collection [105]. Inadequate storage conditions and vulnerable water storage containers, such as those without covers, may serve as a common source for both waterborne and water-related vector-borne diseases [106,107]. In addition to all the problems discussed above, in the case of the school in Vhembe District (South Africa), the school community had to survive for days without water while waiting for the next delivery [96]. Nevertheless, schools with an improved-type water source are less likely to present *E. coli* contamination in stored drinking water compared with schools with unimproved-type water sources [107].

The same discussion also applies to the concept of “improved” sanitation facilities, which considers the existence of an infrastructure designed to hygienically separate excreta from human contact, however, does not contemplate the quality or accessibility of the facilities [34]. Accounting only for the facilities that are kept unlocked, are functional and private (with doors and floors), the ratio of students to sanitation facilities rises significantly [28,42,68,87]. Moreover, the number of schools with “unknown” sanitation facilities across studies (i.e., that could not fit the JMP classification) was very close to the number of schools with improved sanitation facilities (29%, n = 4836 vs. 31%, n = 5201). As can be seen in Table 3, only nine studies deliberated employed the JMP classification. The most frequent type of sanitation facility mentioned in the “unknown” category was pit latrines with no further description of the infrastructure, which hindered the classification of the facility as improved or unimproved. Even though the classification of schools by service levels (basic, limited or no service) would have been a more reliable approach, that was not possible due to the lack of detailed information and standardization among studies.

Only two studies conducted in Ethiopia [45,94] informed the schools’ water consumption per capita per day, which was inferior to the 5 L amount recommended by WHO/UNICEF for students and staff in day schools [103]. It is worth noticing that this threshold covers drinking water requirements and water for personal hygiene, food preparation, cleaning, and laundry [103]. Most likely, the studies did not include this information due to the difficulty in measuring or estimating the quantity of water consumed. However, when provided in the studies, the description of the water supply also points out that the frequency of supply and available storage capacity in schools was insufficient to provide the basic minimum amount of water for the school community [29,62,63,71,78,80,82,87,96,100]. In some cases, students were expected to bring their own water from home, which precludes the schoolchildren from accomplishing the water threshold recommended [62,63,71,78,80,82,100]. The students, especially younger children and those who need to walk long distances to reach school, do not have the physical condition to carry 5 L of water every day. In addition, asking the students to bring water from home means to assume that they have access to that resource at the household level, which may also not be true. Children are one of the population groups more vulnerable to dehydration, which compromises students’ cognitive skills and consequent academic performance [26,108,109]. Furthermore, the long distances between schools and the water sources reported in the studies [66,78,83,91] may decrease the student’s water intake, resulting in dehydration of the children even in schools with on-site water supplies or supplies nearby. The great distances also result in students spending more time out of class in the schools where they were responsible for collecting and carrying water [56,78,89].

Schoolgirls from rural schools in the Shiselweni region (Eswatini) [78] were expected to collect and transport 20 L of water from the river located 2–3 km from the school. The burden of water carrying, which is placed upon the students, can cause injuries, physical pain and stress [110,111]. Moreover, schools with water sources located within 30 min for collection are also less likely to have their drinking water contaminated with *E. coli* as compared to schools with more distant water sources [107]. 

Regarding the content of accessibility to sanitation facilities, the distance from the facilities in schools was not mentioned as a problem, but rather the quality of the pathways often reported as dirty, not well maintained and with several physical barriers, especially for students with disabilities [28,49,64,101,102]. 

Results indicate a general lack of sanitation facilities in schools. None of the studies presented a ratio of girls per sanitation facility in schools that met the WHO/UNICEF standard of 1:25 [103]. As for the boys, 46% (six out of thirteen papers that informed the ratio) satisfied the recommendation of one facility per fifty boys [103]. Even though the guidelines also recommend an additional one urinal for every sanitation facility for boys, urinals were only mentioned in four studies [43,44,76,102] and not in enough quantity. The provision of urinals, which are cheaper and more durable than toilets and latrines [112], could be the solution for several of the problems mentioned in the schools described in the studies. Such a solution can reduce toilet quells, prevent maintenance problems (e.g., clogging or collapse of the infrastructure), and the associated costs, thus improving the longevity of sanitation facilities [112]. Furthermore, it decreases the shared use of facilities between boys and girls and students and teachers. Nevertheless, the sanitation facilities described in the studies were dirty, not properly used and in poor infrastructural conditions, which led to frequent open defecation and urination, putting the students at risk of water-related infections. 

The conditions described in the studies show that the educational institutions were neither girl-friendly nor gender-equity environments. Schools failed to provide three out of the five basic resources and preconditions recommended by the WHO/UNICEF for MHM [38], namely: (i) access to menstrual hygiene materials to absorb or collect menstrual blood; (ii) access to facilities that provide privacy for changing materials and washing body with soap and water; (iii) access to disposal facilities for used menstrual materials (from collection point to final disposal). That, with the addition of the lack of safety and privacy in the sanitation facilities, resulted in girls avoiding going to school during their periods [42,49,51,54,58], and while at school, delaying, not changing or not using the sanitary materials at all [54,70,73], not washing themselves and their menstrual hygiene materials [56,58]. Consequences of poor MHM include toxic shock syndrome [113] and reproductive tract infections [114]. 

Comparing the WASH conditions according to the studies’ settings, the results indicate a lack of research and data from Latin America. From the three papers included [52,71,75], only one study was performed in South America [75], and the information provided for water and hygiene was self-reported by the students and, therefore, not included in the analysis. Moreover, only one study conducted in Central America (Nicaragua) informed the water supply in the institutions, and none of the schools presented improved water sources [71]. The latest WHO/UNICEF report on the progress on WASH in schools with a special focus on COVID-19 [34] also pointed out the lack of sufficient data, especially regarding drinking water, to assess the situation of schools in Latin America and the Caribbean. Even when data was available it was non-representative (e.g., comprised a small portion of the countries that composed that region). 

The schools located in Africa were reported to have better water conditions (higher number and frequency of schools with improved water sources). In terms of sanitation, however, the continent presented the lowest frequency of schools with improved sanitation facilities, which could be the result of the fact that 4726 African schools were classified with unknown sanitation facilities. Compared to what was informed on the latest WHO/UNICEF report [34], the findings of this systematic review indicate a better scenario of drinking water and the worst for sanitation in schools in African countries. While the report states that between 2015 and 2019 the percentage of schools with unimproved or no water source in Sub-Saharan Africa ranged from 46 to 41%, our results indicate that 30% of the schools located in Africa had unimproved or no water source. As for sanitation, the report indicates that 30 to 27% of the schools in Sub-Saharan Africa had unimproved or no sanitation facilities from 2015 to 2019. The results of this systematic review contrast with 49% of schools located in African countries with unimproved or no sanitation facilities. It is noteworthy, however, that the indicators of the report represent the data from 55 Sub-Saharan African countries, whereas, on the other hand, this paper presents the data of schools located in 16 Sub-Saharan African countries. One of the reasons that might explain why the majority of the studies included in this review took place in Africa (53%), and more specifically Sub-Saharan Africa, is probably their known record of water scarcity associated with the location of the Sahara desert. 

Even though Asia was the continent with the highest frequency of schools with improved sanitation facilities, the Asian schools were also the ones with the highest number of students (boys and girls) and girls per sanitation facility ratio. That result indicates that the type of facility is not the issue but rather the quantity available. While the WHO/UNICEF report [34] splits Asian countries into three categories (Central Asia and Southern Asia, Eastern Asia and South-Eastern Asia, and Northern Africa and Western Asia) in this review all Asian countries were aggregated in only one category. Hence, the findings of the review and the report could be directly compared. The report, however, indicates that schools located in Asian countries (in all three categories) have better water and sanitation conditions than the schools in Sub-Saharan African countries [34], while our study finds the opposite trend with regard to drinking water. 

With regard to hygiene, the Asian continent had both the highest frequency of schools with handwashing facilities (82%, n = 372) and the lowest frequency of schools with soap (15%, n = 218). In contrast, the African continent presented the highest number and frequency of schools with soap (52%, n = 348). 

Overall, the findings indicate a general lack of WASH conditions and violation of the HRTWS and its normative contents in schools in LMICs from all regions of the world. The availability of services in schools, both for water and sanitation, was the content most frequently evaluated in the papers (mentioned in 98% of the studies for water, and 88% for sanitation). For drinking water, accessibility was the second most reported content (in 20% of studies), followed by quality and safety (18%), affordability (8%), and last acceptability, dignity and privacy (6%). As for sanitation, quality and safety was the second most mentioned content (60%), followed by acceptability, dignity and privacy (51%), accessibility and affordability tied with 12%. Those results prove the misleading assumption that the sole presence of water and sanitation infrastructure is enough to guarantee its benefits. As evidenced by McMichael (2019) [31], implementation of WASH interventions that only tackle the availability of resources without being accompanied by other contents (such as quality and safety, acceptability, dignity and privacy) might not lead to the expected health and educational benefits. Availability of water can, for instance, reduce transmission of water-washed diseases (such as COVID-19) by enhancing handwashing. If not provided with quality and safety, however, the resource can also intensify waterborne infections. The low percentages of the contents of accessibility, affordability and acceptability, dignity and privacy, especially for water, are of major concern. The lack of WASH disproportionally impacts certain groups of people. By not addressing these contents, the WASH infrastructure present in schools does not consider the vulnerabilities and special needs of social minorities, such as disabled people and women. Schools are important places to reduce the burden of disease related to water supply and sanitation services as they are places where children and adolescents spend a large portion of their day [115,116]. Hence, in order to safeguard the health of all, WASH services in schools should be appropriate and inclusive for all different students. 

### 4.2. The Implications for COVID-19 Pandemic on WASH in Schools

Information extracted from the 65 papers included in the study indicates several obstacles in all WASH domains that hamper schools from implementing the WHO recommended protective measures to minimize and prevent the spread of various infectious diseases, including SARS-CoV-2, among school communities [17]. Hand hygiene is fundamental for the combat of COVID-19. However, handwashing cannot be practiced in schools without a water source or in schools with water available but not for hygiene purposes. The number of schools with water available for handwashing or inside the sanitation facilities was smaller than the number of schools with any type of water source (improved and unimproved), showing that the existence of the water supply in the school does not guarantee its use for hygiene practices. Moreover, the schools reported they had an intermittent water supply [29,62,63,71,78,80,82,87,96,100]. In the face of water scarcity, the resource has to be prioritized for drinking purposes over hygiene. Lack of water availability was the most frequently cited reason for not using handwashing facilities in schools by students in Indonesia [72]. Moreover, it was also mentioned as one of the reasons for not washing hands by the students in Bogotá (Colombia) [75], in Belitung district (Indonesia) [86], and in the Ikenne Local Government Area of Ogun State (Nigeria) [63]. On that note, the implementation of tippy-taps, which as shown in this review, are already becoming part of the infrastructure of schools in LMICs [43,44], is a potentially cheap, quick and easy solution to reconcile water rationing and hygiene practices. Tippy-taps are simple handwashing stations usually constructed with locally available materials that require a very small amount of water (50 mls of water vs. 500 mls of water using tap water) [103,117]. An example of tippy-tappy would be a plastic bottle hung on a rope that pours a small stream of water when it is tipped [103]. Moreover, the design of the station minimizes contact with surfaces, thus, avoiding contamination and further spread of SARS-CoV-2. 

The WHO checklist includes identifying and placing hand hygiene equipment in classroom entrances, on all floors, toilets and canteen facilities [17]. However, according to the reviewed studies, the location of handwashing facilities does not follow a pattern and does not comprise all positions requested. The small number of handwashing stations in the schools might result in the congestion of the facilities and hinder physical distancing between students. The long queues may discourage students from washing their hands or pressure them to be faster, which in turn could lead to suboptimal handwashing and result in suboptimal hand hygiene and cleanliness, a problem that was already reported in the schools before the pandemic [46,94]. The WHO recommends that handwashing with water and soap should be done for at least 40 s and in the best scenario for 60 s [118]. The literature indicates that handwashing with water alone reduces bacteria contamination on hands, and thus, the risk of water-related illness such as diarrhea [119,120]. Notwithstanding, in the case of COVID-19, the presence of the soap is imperative. Due to its enveloped property, human coronaviruses pose low resistance to disinfectants [121,122,123]. Soap and other disinfectants efficiently inactivate the virus by interfering with its lipid envelope, which is essential for preventing infection [121,122,123]. Keeping that in mind, the WHO checklist recommends that schools should ensure adequate and sufficient supplies of soap and hand sanitizer [17]. However, only 19% of the schools (n = 985) addressed in this review could comply with the WHO requirement regarding the presence of soap (not necessarily placed in the handwashing facilities). A lack of soap was the most frequent complaint across studies concerning hygiene (60%, n = 39), being mentioned even in schools with the reported presence of soap. In addition, in some schools in the Sunsari district (Nepal) [90], Dhaka metropolitan and the Manikganj district (Bangladesh) [70], soap was exclusively available for teachers, which also indicates that even when provided, the supply of soap was not sufficient. 

Hand sanitizers such as alcohol-based lotions may be an attractive option for schools in which access to both water and soap is limited, but otherwise should not be considered as the replacement for handwashing. On top of the chemical surfactant action of the soap on the virus, handwashing physically removes the virus by the friction caused by scrubbing [121], which is usually not performed when applying hand sanitizers [123]. Moreover, due to its inflammability and possible child intoxication by ingestion [124,125], hand sanitizers such as alcohol-based type might not be a safe option in the school environment, especially in schools for younger children such as primary, pre-school and daycare institutions. Even though both hygiene resources are effective against COVID-19, considering that shortage of funds was the limitation reported throughout the papers for the lack of soap in schools, soap has the advantages of being easier and cheaper to access [118,126], which makes handwashing a more equitable solution.

As the last alternative, the WHO brief offering interim guidance on WASH for COVID-19 management suggests that ash, the residue from stoves and fires, should be used for hand cleaning when other resources (soap, water and alcohol-based locations) are not available [127]. In low-income communities, especially in countries in the Asian and African region such as India, Bangladesh, Pakistan, and sub-Saharan countries, ash is often used for post-defecation hand hygiene [128]. However, the evidence of the benefits of hand cleaning with ash for reducing the spread of viral and bacterial infections is limited [129]. Hitherto, no study was found linking hand hygiene practices with ash as an effective measure to prevent COVID-19. Further research should be conducted to evaluate if the benefits of hand hygiene with ash surpass its potential risks, including the transmission of gastrointestinal, parasitic and other infectious diseases and contamination with toxic metallic compounds [128]. In the case of ash, if proven to be an effective and safe alternative material for hand hygiene, given its low cost, its use could have a major impact in Asian and African countries, where the practice is common and schools lack water, soap and funding to purchase hand cleaning resources.

The relevance of hand hygiene practices (handwashing with soap and water, hand sanitizers or even with ash) in the context of the COVID-19 pandemic is not restricted to the school environment. However, when attending school, children are still learning the basic principles of hygiene and often have the behavior of touching each other frequently, surfaces and putting items and their hands, mouth and noses [130]. Acquisition of hygiene habits in childhood is crucial as those are the habits that will most likely prevail for the rest of their life. Habits take time to establish and need repeated practice to become a routine [131,132]. Schools can help in this process through the reinforcement of ritualization and normalization of hygiene habits [131,132]. Moreover, even though “long covid” (e.g., manifestation of persistent long-term symptoms after the acute infection) has been more frequently reported among adult patients, in children the disease could affect their cognitive development, which has been already compromised during the pandemic due to delayed education, isolation, increase in poverty and food security, among other factors [133,134]. In that sense, the provision of hygiene infrastructure in schools should not be overlooked. As stated by Mushi and Shao (2020) [135], WASH services will function as a mechanism in mitigating secondary effects of COVID-19 during the subsequent recovery phases. In the specific case of the school environment, the improvement of WASH services will prevent that education and development of children and adolescents to be further delayed due to the spread of other infectious diseases and will promote a clean, healthy and secure space for learning.

Even though the fecal-oral route has not yet been proven to be one of the transmissions pathways for SARS-CoV-2, the water and sanitation conditions in the schools described by the studies pose an extra risk for the school community. Keeping in mind that the virus can remain viable for days in raw sewage, sewage sludge and feces of patients [23], the inadequate disposal of the sewage, such as in stagnant pools around the school premises [61,76] represents an additional exposure route. Without the proper sewage treatment and subsequent disposal, the water sources in the schools are at risk of getting contaminated with the virus, thus, enabling waterborne transmission (i.e., by the ingestion of SARS-CoV-2) [136]. On that note, it is important to highlight how the monitoring of sanitation in non-household settings, such as in schools, has been pushed aside in favor of household settings [104]. While the JMP Sanitation ladder for households accounts for the safe disposition and treatment of sewage, the same is not included in the JMP schools WASH ladder [34,137].

The contaminated water, if used for cleaning purposes, could also contribute to the water-washed transmission by spreading the virus to surfaces [138]. Water disinfection techniques could effectively prevent both transmission routes. However, only a small portion of the schools (33%, or 1003 out of 3065) in nine countries reported that the water at the school was treated before consumption or chlorine residual was identified in the water. Moreover, the wastewater attracts insect vectors that may carry the virus, further contaminating additional surfaces [138]. 

The use of shared sanitation facilities in schools, along with its unhealthy state with visible contamination of feces, put students at risk of water-washed COVID-19 transmission (i.e., touch their mouths, noses or eyes with contaminated hands) [139,140]. This risk is even higher for three different groups of students. First, women due to their more frequent use of facilities during menstruation. Second, disabled students who attend schools without disability-friendly bathrooms and need to touch the ground to access the facilities [102]. Last, for children and adolescents that attend schools where students are in charge of cleaning the toilets, such as those reported in Trapeang Chour Commune (Cambodia) [80], Ibadan (Nigeria) [62] and in Rumphi (Malawi) [102]. The decontamination of the environment, as stated by the WHO checklist [17], with special attention to water and sanitation facilities, however, cannot be feasible considering that schools also lack cleaning supplies [43,54,80,95]. When practicing open defecation, which was frequently reported in the studies, students may not clean their hands afterward [63]. Lack of self-cleaning materials, as described in the schools, might also discourage and hamper students from applying hygiene practices. As shown by Shehmolo et al. (2021) [94], the state of water and sanitation conditions in schools are crudely associated with the level of hygiene practices of the students. Therefore, regardless of whether the fecal-oral hypothesis is confirmed, prevention of COVID-19 relies on the provision of adequate water and sanitation conditions in schools, which will boost handwashing and hygiene etiquette in the environment.

## 5. Limitations of the Review

Some limitations of this review have to be highlighted. First, this review could not capture WASH in schools in Latin America, which led to the underrepresentation of this geographical region. Second, results of this review need to also be contextualized with a reporting bias from schools that are accessible to researchers and international organizations reporting on them—thus, probably, the schools investigated in the included studies have already higher WASH standards than schools in areas “outside of reach or interest” of stakeholders (e.g., researchers, organizations, etc.). Third, WASH conditions in schools were not analyzed under the context of locality (rural vs. urban) or management model (public vs. private), factors that are known to interfere in schools’ infrastructure. Forth, despite being another component of WASH that is linked to health-related outcomes, solid waste management and disposition in schools was not part of the scope of this research. Last, screening of titles, abstracts and full-texts against inclusion criteria, data extraction and quality assessment were carried out by only one researcher. The involvement of at least two independent reviewers would have enhanced the quality of this review.

## 6. Conclusions and Future Research

Results of this systematic review on WASH in schools in LMICs, considering 18,465 schools described in 65 studies across 30 different countries, indicate a lack of appropriate WASH and MHM conditions in schools in all continents. Even when infrastructure was present in the schools, its quantity was not enough, and its state was inappropriate to attend and safeguard the school community’s health. The human right to safe drinking water and sanitation and its normative contents (availability, accessibility, affordability, quality and safety, acceptability, privacy and dignity) have not been guaranteed in schools. Even though most of the schools reported having an improved water source, water intermittency, long distances from the water source, contamination of water, problems with maintenance of the water supply and lack of funding for water provision in schools were frequently mentioned. As for sanitation, most of the facilities described in schools could not be classified as improved or unimproved following the JMP definitions. The majority of schools could not comply with the WHO/UNICEF sanitation facility ratio standards. Facilities were dirty, not properly used and in poor infrastructural conditions, which led to frequent open defecation and urination. In the context of the COVID-19 pandemic, the current school’s infrastructure hampers students to practice handwashing, a fundamental strategy to contain the spread of SARS-CoV-2 among school communities. 

Even though the fecal-oral route has not yet been proven to be one of the transmissions pathways for SARS-CoV-2, the provision of WASH services in schools is essential for the prevention of COVID-19 and other water-related diseases in the school environment. Based on the results of this systematic review we recommend that further research should be conducted to:Assess what is the situation of WASH conditions in schools in Latin America;Describe the differences of WASH conditions in schools in LMICs according to the locality (rural vs. urban) and model of management (public vs. private);Identify what WASH interventions have been implemented in schools during the ongoing COVID-19 pandemic in order to provide safe reopening and how they satisfy the normative contents of HRTWS;Explore emergent themes in the school environment, such as MHM, gender discrimination and inequalities, and disability-friendly WASH services;Discuss how to improve standardization across studies (e.g., enhance the use of the JMP service ladders and definitions) in order to allow the comparison of WASH services in different locations;Investigate how to integrate the normative contents of the HRTWS to the JMP service ladders and definitions.

## Figures and Tables

**Figure 1 ijerph-19-03124-f001:**
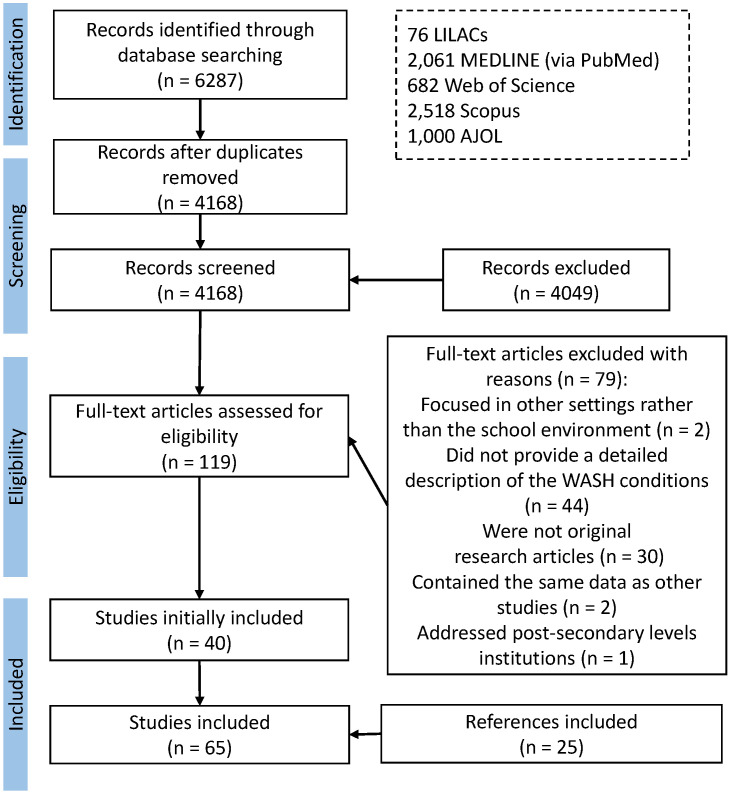
PRISMA flow diagram for the systematic literature review on water, sanitation and hygiene in schools in low- and middle-income countries.

**Figure 2 ijerph-19-03124-f002:**
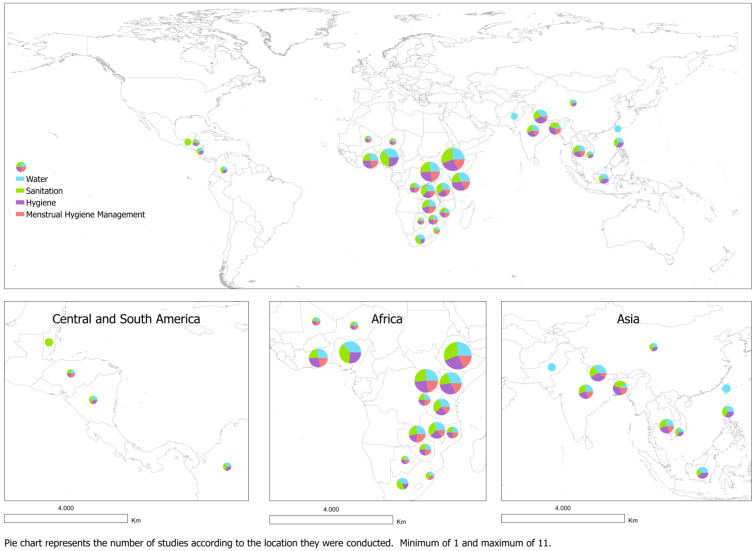
Distribution of the 65 publications included in the systematic review according to the location the studies were conducted.

**Figure 3 ijerph-19-03124-f003:**
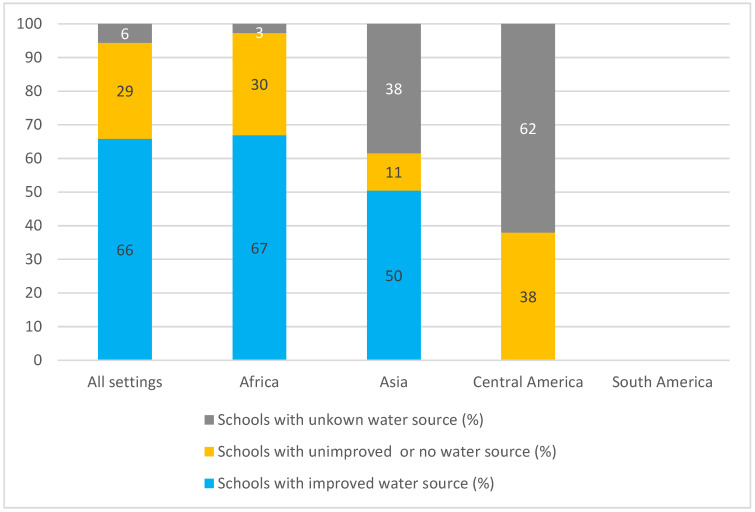
Percentage of schools with improved, unimproved/no drinking water source and unknown source based on the review of the literature (n = 65 publications). Number of schools in each setting: Africa n = 12,997; Asia n = 832; Central America = 456; South America n = 0; All settings n = 16,963. The total number of schools assessed in all settings (16,963) is lower than the sum of schools per continent (14,285) as some of the studies did not identify the schools per location.

**Figure 4 ijerph-19-03124-f004:**
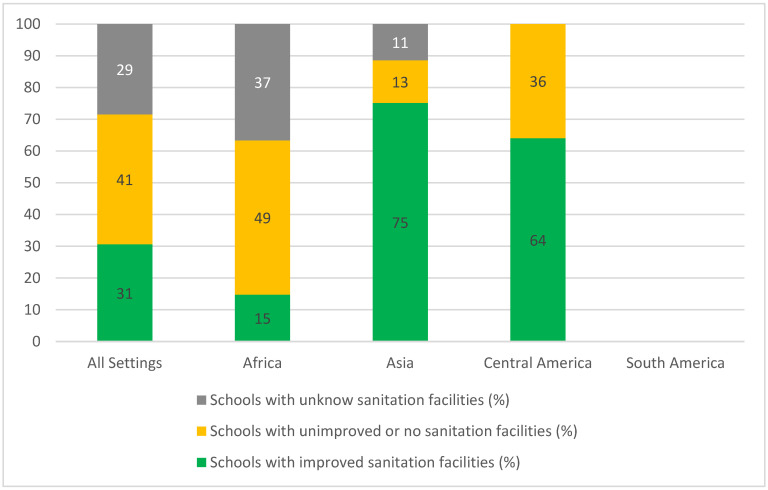
Percentage of schools with improved, unimproved/without and unknown sanitation facilities based on a review of the literature (n = 65 publications). Number of schools in each setting: Africa n = 12,897; Asia n = 961; Central America = 412; South America n = 0; All settings n = 16,960. The total number of schools assessed (16,960) is lower than the sum of schools per continent (14,270) as some of the studies did not identify the schools per location.

**Figure 5 ijerph-19-03124-f005:**
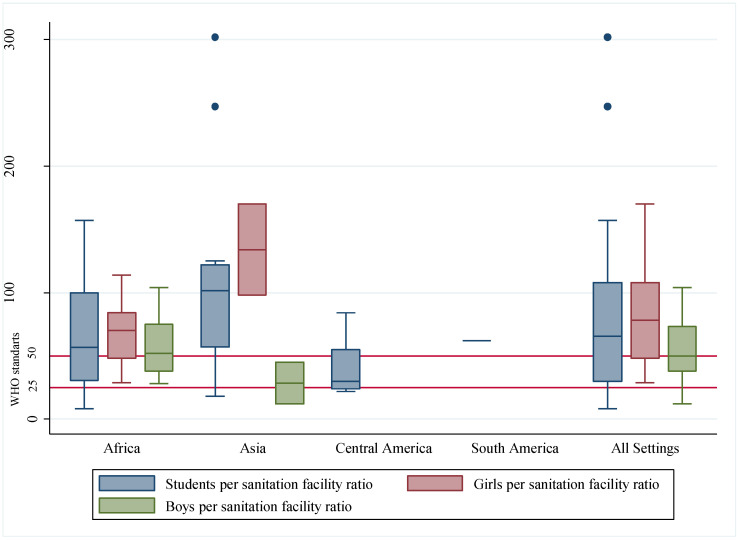
Distribution of students (boy and girls), boy and girl per sanitation facility ratios in schools based on a review of the literature (n = 65 publications). Reference lines: WHO guidelines of one facility per 25 girls and per 50 boys.

**Figure 6 ijerph-19-03124-f006:**
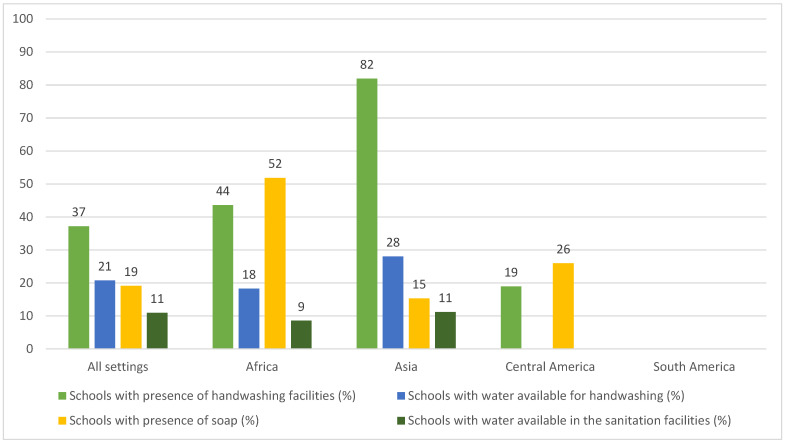
Percentage of schools with hygiene and related facilities according to the location of studies’ settings based on a review of the literature (n = 65 publications). Number of schools in each setting assessed for handwashing facilities: Africa n = 2629; Asia n = 454; Central America n = 464; South America n = 0; All settings n = 6237. Number of schools in each setting assessed for water available for handwashing: Africa n = 2610; Asia n = 100; Central America = 0; South America n = 0; All settings n = 5400. Number of schools in each setting assessed for the presence of soap: Africa n = 671; Asia n = 1422; Central America n = 362; South America n = 0; All settings n = 5151. Number of schools in each setting assessed for water available in the sanitation facilities: Africa n = 105; Asia n = 775; Central America n = 0; South America n = 0; All settings n = 886. The total number of schools assessed is lower than the sum of schools per continent as some of the studies did not identify the schools per location.

**Table 1 ijerph-19-03124-t001:** Exclusion criteria for the systematic literature review on WASH in schools in low- and middle-income countries.

Exclusion Criteria	Sub-Criteria
Lack of detailed description of WASH conditions	The theme of the study was right, however, the study did not present a detailed description of the WASH conditions
Wrong location	Not in low- and middle-income countries
Wrong educational level	Addressed universities, faculties and colleges
Wrong setting	Focused on other settings rather than the school environment (e.g., household, healthcare facilities, etc.)
Wrong study type	Did not present original research (e.g., systematic reviews, study protocol, short communication, etc.)
Grey literature	-
Duplication of information	Paper partially contained the same information included in other publications (in case of publications from the same research group)

**Table 2 ijerph-19-03124-t002:** Description of extracted data.

Topics	Description of Extracted Data
Publication	Reference (authors and year of publication), year of data collection and location where the study was conducted
School	Type of educational institution, number of schools, locality (urban vs. rural) and management model (private vs. public)
Thematic addressed	Components of WASH and MHM that were addressed in the studies, and with specific regards to water and sanitation the normative contents of the HRTWS that were mentioned
Water	Drinking water-related specifics according to the normative contents of the HMRTWS, schools with improved water source, schools with unimproved or no water source, schools with “unknown” water source, ratio of water tap to school population, reported water shortage, and reported maintenance problems with water supply in the schools
Sanitation	Sanitation facility-related specifics according to the normative contents of the HMRTWS, schools with improved sanitation facilities, schools with unimproved or no sanitation facilities, schools with “unknown” sanitation facilities, students per sanitation facility ratio, girls per sanitation facility ratio, boys per sanitation facility ratio, reported lack of cleanliness, reported shared facilities between boys and girls, reported shared facilities between students and teachers, reported lack of doors, reported lack of locks and reported lack of roofs
Hygiene	Schools with the presence of handwashing facilities, type, number and location of handwashing stations, student-to-handwashing basin ratio, schools with water available for handwashing, schools with the presence of soap, schools with water available in the sanitation facilities, reported lack of handwashing facilities, reported lack of soap, reported lack of water in the sanitation facilities, reported lack of anal cleaning materials/self-cleaning, reported lack of cleaning materials
MHM	Reported lack of access to menstrual hygiene materials to absorb or collect menstrual blood, reported lack of disposal facilities for used menstrual materials such as bins and trash cans for sanitary materials disposal, reported lack of room for changing, bathing, or washing sanitary materials

**Table 3 ijerph-19-03124-t003:** Description of the 65 studies included in the systematic review.

Study	Country	Type of School	Number of Schools	Water					Sanitation					Hygiene	MHM	JMP Definitions
				1	2	3	4	5	1	2	3	4	5			
Agol and Harvey, 2018 [27]	Zambia	-	10,000	X	X				X						X	X
Ahmed et al., 2020 [41]	Pakistan	Primary	425	X			X									X
Alam et al., 2017 [42]	Bangladesh	Primary and Secondary	700						X					X	X	X
Alexander et al., 2014 [43]	Kenya	Primary	62	X					X		X	X	X	X	X	
Antwi-Agyei et al., 2017 [44]	Tanzania	Primary	70	X		X			X		X	X		X		X
Aschale et al., 2021 [45]	Ethiopia	Primary	5	X			X		X			X	X	X		
Assefa and Kumie, 2014 [46]	Ethiopia	Primary	5	X					X	X		X	X	X		
Babalobi, 2013 [47]	Nigeria	Primary	4	X		X		X	X							
Bergenfeld, Jackson and Yount, 2021 [48]	Nepal	Secondary	159	X					X					X	X	
Boosey, Prestwich and Deave, 2014 [49]	Uganda	Primary	6	X					X	X		X	X	X	X	
Bowen et al., 2007 [50]	China	Primary	87	X					X					X		
Bulto, 2021 [51]	Ethiopia	Preparatory and High	3												X	
Chatterley, Liden and Javernick-Will, 2013 [52]	Belize	Primary School	15						X			X				
Chatterley et al., 2014 [53]	Bangladesh	Primary School	16						X		X	X		X		
Chinyama et al., 2019 [54]	Zambia	Primary and Secondary	6	X					X		X		X	X	X	
Chung et al., 2009 [55]	Taiwan	Above senior high school level and schools under junior high school level	42	X			X									
Connolly and Sommer, 2013 [28]	Cambodia	Secondary	2	X					X	X		X	X	X	X	
Crofts and Fisher, 2012 [56]	Uganda	Secondary	18	X	X							X	X	X	X	
Cronk et al., 2021 [57]	Ethiopia, Ghana, Honduras, India, Kenya, Malawi, Mali, Mozambique, Niger, Rwanda, Tanzania, Uganda, Zambia, and Zimbabwe	Primary and Secondary	2690	X	X		X		X			X		X	X	X
Degefu Birhane, Serbessa and Degfie, 2019 [58]	Ethiopia	Junior	5	X									X		X	
Devkota et al., 2020 [59]	Nepal	-	1	X										X		
Dube and January, 2012 [60]	Zimbabwe	Primary	4	X					X			X		X	X	
Ebong, 1994 [61]	Nigeria	Secondary	1	X					X			X	X			
Egbinola and Amanambu, 2015 [62]	Nigeria	Secondary	44	X	X				X	X		X	X	X		
Ekpo et al., 2008 [63]	Nigeria	Primary	3	X					X			X	X	X		
Erhard et al., 2013 [64]	Uganda and Malawi	Primary	41	X	X			X	X	X		X		X		
Ezeonu and Anyansi, 2010 [65]	Nigeria	Primary	31	X	X				X			X		X		
Freeman et al., 2014 [66]	Kenya	Primary	185	X			X		X					X		X
Grant, Lloyd and Mensch, 2013 [67]	Malawi	Primary	59	X					X			X	X		X	
Grimes et al., 2017 [68]	Ethiopia	Primary	30						X			X		X		
Hassen and Abera, 2015 [69]	Ethiopia	Primary and Secondary	10	X		X			X				X	X		
Jahan et al., 2020 [70]	Bangladesh	-	8	X					X			X	X	X	X	
Jordanova et al., 2015 [71]	Nicaragua	Pre-school, Primary, Secondary, with all levels and unspecific schools	526	X	X		X		X		X	X	X	X		
Karon et al., 2017 [72]	Indonesia	Primary and combined Primary and Junior high	75	X	X	X		X	X		X	X	X	X		X
Korir, Okwara and Okumbe, 2018 [73]	Kenya	Primary	10	X					X			X	X	X	X	
Lang, 2012 [74]	Ghana	Elementary	4	X										X		
Lopez-Quintero, Freeman and Neumark, 2009 [75]	Colombia	-	25	X					X					X		
Majra and Gur, 2010 [76]	India	Primary, Upper Primary and from Primary to High school level	20	X					X			X	X	X		
Mathew et al., 2009 [77]	India	Upper Primary	300	X	X				X			X	X	X	X	
Mbatha, 2011 [78]	Eswatini	Primary	2	X	X		X		X			X			X	
Miiro et al., 2018 [79]	Uganda	Secondary	4	X					X					X	X	
Mirassou-Wolf et al., 2017 [80]	Cambodia	Primary and Secondary	8	X			X		X			X	X	X		
Mogaji et al., 2016 [81]	Nigeria	Primary School	3	X					X			X	X	X		
Mohammed and Larsen-Reindor, 2020 [82]	Ghana	Junior High	5	X					X	X			X	X	X	
Montgomery et al., 2016 [83]	Uganda	Primary	8	X	X				X					X	X	
Morgan et al., 2017 [84]	Ethiopia, Kenya, Mozambique, Rwanda, Uganda, and Zambia	Primary, Secondary and combined schools	2270	X	X		X		X					X	X	X
Mwanri, Worsley and Masika., 2000 [85]	Tanzania	-	76	X					X			X	X			
Nazliansyah, Wichaikull and Wetasin, 2016 [86]	Indonesia	Elementary	11	X										X		
Ngwenya et al., 2018 [87]	Botswana	Primary	3	X					X			X	X	X	X	
Ofovwe and Ofili, 2009 [88]	Nigeria	Primary	133	X					X					X		
Parker et al., 2014 [89]	Uganda	Primary and Secondary	14	X	X				X			X	X	X	X	
Rai et al., 2017 [90]	Nepal	-	40	X					X			X	X	X		
Saboori et al., 2011 [91]	Kenya	Primary	55	X	X	X	X							X		
Sangalang et al., 2020 [92]	Philippines	Primary and Secondary	15	X			X		X			X		X		X
Shallo, Willi and Abubeker, 2020 [93]	Ethiopia	High	5	X								X		X	X	
Shehmolo et al., 2021 [94]	Ethiopia	Primary	8	X	X				X				X	X		
Shrestha et al., 2017 [95]	Nepal	Secondary or above	16	X			X		X		X	X	X	X		
Sibiya and Gumbo, 2013 [96]	South Africa	Secondary	8	X					X					X		
Sommer et al., 2015 [29]	Ghana, Cambodia and Ethiopia	Secondary	6	X					X			X	X	X	X	
Sommer, 2013 [97]	Tanzania	Primary, Secondary and Boarding schools	12	X					X				X	X	X	
Uduku, 2015 [98]	Ghana and South Africa	Primary	2	X					X			X				
Vally et al., 2019 [99]	Philippines	Elementary	8	X	X				X			X	X	X		
Wichaidit et al., 2019 [100]	Kenya	Primary	30	X										X		
Xuan et al., 2012 [101]	Vietnam	Primary and Secondary	6	X					X	X	X	X	X	X		
Zaunda et al., 2018 [102]	Malawi	Primary	10	X	X			X	X	X		X	X	X		

The number below water and sanitation refer to the normative contents of the HRTWS: 1-Availability; 2-Accessibility; 3-Affordability; 4-Quality and safety; 5-Acceptability, privacy, and dignity.

**Table 4 ijerph-19-03124-t004:** Water quality reported in schools based on a review of the literature (n = 65 publications).

Study	Country	
Chung et al., 2009 [55]	Taiwan	26% of schools (11 out of 42) had the water samples in non-conformity with the national standards for water quality
Sangalang et al., 2020 [92]	Philippines	20% of schools (3 out of 15) had water that was contaminated by *E. coli*
Shrestha et al., 2017 [95]	Nepal	75% of school drinking water source samples and 76.9% point-of-use samples (water bottles) collected in 16 schools were contaminated with thermo-tolerant coliforms
Ahmed et al., 2020 [41]	Pakistan	Drinking-water samples collected in 425 schools were contaminated with *E. coli* (49%), *Salmonella* spp. (54%), *V. cholerae* (49%) and *Shigella* (63%), respectively
Morgan et al., 2017 [84]	Ethiopia, Kenya, Mozambique, Rwanda, Uganda, and Zambia	No rural schools in Mozambique (n = 198), Zambia (n = 576) and Uganda (n = 251) had very high-risk water quality. Most of the rural schools in all the countries assessed had samples with *E. coli* counts in the lowest risk category (79% considering only water samples taken from the source and 77% considering water samples taken from stored water).
Cronk et al., 2021 [57]	Ethiopia, Ghana, Honduras, India, Kenya, Malawi, Mali, Mozambique, Niger, Rwanda, Tanzania, Uganda, Zambia, and Zimbabwe	Zambia had the highest proportion (80%) of schools with water that conformed with the WHO guideline value for *E. coli*, while Honduras (22%) and Tanzania (16%) had the lowest compliance

**Table 5 ijerph-19-03124-t005:** National requirements of students per sanitation facilities found across studies based on a review of the literature (n = 65 publications).

	Girls per Sanitation Facility	Boys per Sanitation Facility	Students per Sanitation Facility
Kenya [43,73]	25:1	30:1	-
Philippines [92]	Two toilets for 30–100 female students with an increment of one toilet for each additional 100 female students	50:2 for 50 or more male students with an increment of one toilet for each additional 100 male students	-
Tanzania [97]	20:1	-	-
Tanzania [44]	40:1	50:1	-
Zambia [27]	-	-	20:1
Colombia [75]	-	-	25:1
WHO [44,85,103]	25:1	50:1	-

## Data Availability

Data are contained within the article and Appendix A.

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
