# Peer review of "Water, Sanitation and Hygiene in Schools in Low- and Middle-Income Countries: A Systematic Review and Implications for the COVID-19 Pandemic"

_ijerph, 2022, doi:10.3390/ijerph19053124_

Round 1

Reviewer 1 Report

This is an important review of WaSH in schools and the implications for COVID. There are general and specific comments below to help improve the review.  

General 

  1. The statement that this is the first systematic review of WaSH in schools in LMICs is misleading. It may be the first review of wash in schools with a reflection on the implications for COVID. There is a reivew has been done that on wash in schools in LICs, and should be acknowledged by C McMichael et al. in the IJERPh in 2019. This article should be reflected on in the discussion as well to mention what more this review adds to the body of literature. https://www.mdpi.com/1660-4601/16/3/359
  2. Search strategy should include the dates searched (stated in the abstract but not the search strategy (publication years included). Also what were the exclusion criteria?
  3. Explaining the inclusion and exclusion criteria at each stage – title, abstract, and full-text would be helpful. There should be some description of how many people were involved at each stage and how discrepancies were dealt with in this review.
  4. The information extracted considers the five normative contents of the human right to water and sanitation – availability, accessibility, affordability, quality, and acceptability /improved vs unimproved. Hygiene is according to JMP definitions. The connection of all of these areas to COVID could be more clearly outlined at the start. The hygiene connection and sewage connection was mentioned. The sharing of facilities and risks of contracting COVID in unsanitary/unhygienic facilities is not sufficiently mentioned. The MHM connection to COVID is also not really stated.
  5. Studies classified according to MMAT mixed Methods Appraisal Tool. Could this analysis be included in the paper and not as a supplement.
  6. The map of where the studies took place is helpful. The preponderance of studies from Africa should be mentioned in the discussion, as this reviews conclusions then are most specific to this region.
  7. As analyzed for the each of the normative content of the human right for water and sanitation (availability, accessibility, affordability, quality, and acceptability), there was the percent of articles that analyzed this factor and the N studies. This should be done consistently, and also be given for sanitation accessibility, affordability, and acceptability.
  8. As this paper is focusing on COVID, providing the results for hygiene, sanitation, and then drinking water might be more appropriate.
  9. In the hygiene section of the results 3.6, the graph should a bar with information on the schools with both water and soap, as without one you don’t have handwashing facilities, as per the JMP definition for schools.
  10. As the focus is on COVID, I think there is a missed opportunity. In the discussion, you could mention how treatment of sewage and safe sewage disposal is not well tracked in the JMP schools WaSH ladder, but is tracked in the household ladder. It would be good to have more information for schools.
  11. Section 3.57 on MHS tells the reader how many articles discussed MHS. It would be good to also discuss how many describe the different MHS available, according to JMPs definition.  The presence of privacy for changing, availability of menstrual hygiene materials, and a private place to wash should all be analyzed to determine if these are present in the articles (N and %) as well as the deficiencies.  

https://washdata.org/monitoring/menstrual-health

  1. The section on quantity in the discussion, should acknowledge the difficulty in measuring water quantity consumed or estimating it.
  2. The implications for COVID should be expanded as this was a main reason for the review. For example – the number of articles/schools that have water and soap present was not summarized in the results and should be discussed here in the discussion.
  3. Further connections to COVID should be made throughout the results section.
  4. In the limitations section, there was a lack of studies from some region on WaSH in schools.
  5. In discussion, the 2020 wash in Schools update from JMP should be mentioned to see if the results are in line with yours, as you are trying to make representative statements about regions. The JMP is more representative and could add to your argument. It also could illuminate more on countries where you have little/no data.
    1. The report should be cited and their representative data given in the discussion to give additional weight to your arguments about a lack of hand hygiene supplies in schools, lack of treatment or adequate disposal of feces, and general lack of soap.
  6. The cleanliness of the sanitation facilities should be further discussed as a COVID risk. These are shared sanitation facilities and there is substantial evidence that they are not cleaned often or well.

Specific

Pg 2 ln 49 severely – word choice or delete or give a rate of increase.

Ln 72-74 break up sentence. It is unclear what is the main point of this sentence.

Lns 91-121 – very long paragraph – suggest breaking it up.

Ln 94 Add “health and educational” :innumerable health and educational benefits.

Ln 102 – delete the between 2010 and access.

Ln 420 – in the table delete “These” after the country list for Cronk’s 2021 study

Ln 619 – the appropriate word is gender equity not equality, as most men don’t menstruate and so the presence of equal facilities is not appropriate. Equity is more appropriate here.  

Ln 648 overstates –“ Although the concept of  “i_m_p_r_o_v_e_d_” _w_a_t_e_r_ _s_o_u_r_c_e_ _i_n_v_o_l_v_e_s_ _t_h_e_ _p_o_t_e_n_t_i_a_l_ _t_o_ _d_e_l_i_v_e_r_ _s_a_f_e_ _w_a_t_e_r_ _[34], the classification 647 itself does not guarantee that the drinking water provided by the schools truly safeguards 648 the health of the school community. ” It does not guarantee that the drinking water provided by the schools is safe against microbiological and chemical contamination.  Safeguarding the health of the community is beyond what safe water can do, much more must be present to “safeguard the health of the community”

Ln 732 – gender equality environments might be better written as WaSH environments that incorporate gender equity or gender equity WaSH environments.

Line 762, does this match what the JMP states for WaSH in Schools? The JMP data has more countries represented and is more representative of each country, given the sampling design. It is difficult to make any conclusions about 1 continent with 1 study. Also, you should summarize the number of studies from Africa and the number from Asia in this paragraph because you are making statements about the entire continent.

Ln 839 need not needs “to touch their hands”

Reviewer 2 Report

Dear Authors,

thank you for the opportunity to review this very valued work that provides a well-rounded systematic review of literature. First of all, let me just briefly state that I enjoyed reading it very much, and as an evidence synthesis methodologist and public health researcher, I appreciate the topic and what it brings. There are many aspects I could praise, where to begin!

The review follows the highest current standard of reporting and it followed an a priori published protocol. The methods are presented clearly, and the review has an excellent standard of language and writing. I am absolutely taken away by how well the introduction, and the whole review, is written. It is clear, it covers all the important themes so anyone can understand what you are doing and why. E X C E L L E N T !

I have a few minor suggestions to take the review just one small step further. Also, I appreciate and applaud that you were able to complete the review in such a short time (considering the search was conducted in April last year).

The two minor suggestions are to use and cite the new PRISMA statement (2020/2021) and to change „PubMed“ to „MEDLINE (via PubMed)“.

I cannot express how much I appreciate how well-conducted your review is. Thank you for an excellent read and the value of the work!

Kind regards

Reviewer 3 Report

General comment:

The manuscript presents an interesting systematic literature review about a relevant topic, that is, the water, sanitation and hygiene (WASH) approach in schools located in low and middle-income countries and its implications for the COVID-19 pandemic. The manuscript is adequately structured, making use of tested and rigorous tools for ensuring the reliability of results. Nevertheless, there are several aspects that deserve further work and refinement, including the theoretical framework, some descriptive information displayed in the figures; as well as the formulation of the research agenda and public health policies.

Specific comments:

In order to improve the global quality of the manuscript, it is provided below a set of comments and recommendations to be considered in the revision process of the manuscript:

  1. The manuscript needs to be revised by a professional editor and English native.
  2. The positioning of the manuscript, in terms of relevance in fostering public health and reducing inequalities in developing countries (that is, low and middle-income countries), as well as the contributions of the manuscript need to be clearly outlined in the introductory item.
  3. The theoretical background needs to be seriously improved using reference studies from your target journal.
  4. In line 240, please check the numbering used for the 5 categories.
  5. Verify the percentages used in Figures 3 and 4.
  6. The research agenda needs to be improved linking the caveats and problematics found in previous studies.
  7. A set of implications for public health and education need to be provided.

Round 2

Reviewer 3 Report

Considering the changes made, it is recommended the acceptance of the manuscript, since the authors addressed my previou comments in a satisfactory way.